# Generative Modeling for Interpretable Anomaly Detection in Medical Imaging: Applications in Failure Detection and Data Curation

**DOI:** 10.3390/bioengineering12101106

**Published:** 2025-10-14

**Authors:** McKell E. Woodland, Mais Altaie, Caleb S. O’Connor, Austin H. Castelo, Olubunmi C. Lebimoyo, Aashish C. Gupta, Joshua P. Yung, Paul E. Kinahan, Clifton D. Fuller, Eugene J. Koay, Bruno C. Odisio, Ankit B. Patel, Kristy K. Brock

**Affiliations:** 1Departments of GI Radiation Oncology, Imaging Physics, Interventional Radiology, and Radiation Oncology, The University of Texas MD Anderson Cancer Center, Houston, TX 77030, USA; 2Departments of Computer Science and Electrical and Computer Engineering, Rice University, Houston, TX 77005, USA; 3Department of Biology, Texas Southern University, Houston, TX 77004, USA; 4Departments of Radiology and Bioengineering, University of Washington, Seattle, WA 98195, USA; 5Department of Neuroscience, Baylor College of Medicine, Houston, TX 77030, USA

**Keywords:** generative modeling, anomaly detection, failure detection, data curation, generative adversarial network

## Abstract

This work aims to leverage generative modeling-based anomaly detection to enhance interpretability in AI failure detection systems and to aid data curation for large repositories. For failure detection interpretability, this retrospective study utilized 3339 CT scans (525 patients), divided patient-wise into training, baseline test, and anomaly (having failure-causing attributes—e.g., needles, ascites) test datasets. For data curation, 112,120 ChestX-ray14 radiographs were used for training and 2036 radiographs from the Medical Imaging and Data Resource Center for testing, categorized as baseline or anomalous based on attribute alignment with ChestX-ray14. StyleGAN2 networks modeled the training distributions. Test images were reconstructed with backpropagation and scored using mean squared error (MSE) and Wasserstein distance (WD). Scores should be high for anomalous images, as StyleGAN2 cannot model unseen attributes. Area under the receiver operating characteristic curve (AUROC) evaluated anomaly detection, i.e., baseline and anomaly dataset differentiation. The proportion of highest-scoring patches containing needles or ascites assessed anomaly localization. Permutation tests determined statistical significance. StyleGAN2 did not reconstruct anomalous attributes (e.g., needles, ascites), enabling the unsupervised detection of these attributes: mean (±standard deviation) AUROCs were 0.86 (±0.13) for failure detection and 0.82 (±0.11) for data curation. 81% (±13%) of the needles and ascites were localized. WD outperformed MSE on CT (*p* < 0.001), while MSE outperformed WD on radiography (*p* < 0.001). Generative models detected anomalous image attributes, demonstrating promise for model failure detection interpretability and large-scale data curation.

## 1. Introduction

Generative artificial intelligence (AI) excels at modeling complex distributions of medical imaging data [1], enabling unsupervised anomaly detection by identifying deviations from learned representations of normality [2]. Among generative approaches, reconstruction-based methods are the most commonly used, particularly those employing generative adversarial networks and variational autoencoders [3]. In medical imaging, these techniques have primarily been used for unsupervised disease detection across a wide range of anatomical regions and imaging modalities [4,5]. 

Despite its success in disease detection, generative modeling-based anomaly detection has seen limited application in important areas of medical imaging. In this study, we expand its use to two novel tasks: (1) enhancing the interpretability of AI model failure detection in clinical settings, and (2) supporting automated quality control for large-scale medical imaging repositories. These tasks address distinct but critical challenges—ensuring safe deployment of AI systems in clinical workflows and improving the scalability of data curation for medical imaging research.

First, we investigate whether generative models can spatially localize anomalies that contribute to AI model failures. AI models often underperform on inputs that differ from their training distributions [6], a challenge exacerbated in medical imaging by limited amounts of annotated data and population heterogeneity [7]. For example, Anderson et al. reported that a liver segmentation model, which performed well in most cases, failed on cases containing metal stents and ascites—features absent from the training data [8]. Detecting such failures is critical for patient safety, particularly for individuals with underrepresented characteristics. While existing failure detection methods can flag problematic inputs [9], they often lack interpretability, which is essential for clinical validation and trust. Our study explores whether generative modeling can improve interpretability by spatially localizing the anomalies responsible for these failures, with a case study focused on unsupervised localization of metal artifacts and ascites in abdominal CT scans—the same anomalies identified in Anderson et al. [8].

Second, we apply generative modeling to support automated data curation in large repositories. These repositories are pivotal for advancing medical AI research by enabling broad access to diverse imaging datasets [10]. However, scaling manual data curation remains a significant challenge [11]. For instance, the Medical Imaging and Data Resource Center (MIDRC) has acquired 573,506 imaging studies, with 374,785 undergoing quality assessment and harmonization as of October 2025 (Medical Imaging and Data Resource Center. https://www.midrc.org/. accessed on 2 October 2025). We propose using generative modeling to detect anomalous submissions, with a case study in identifying MIDRC chest radiographs that deviate from those found in the ChestX-ray14 dataset [12], which serves as a widely used benchmark for chest imaging.

We hypothesize that generative modeling can spatially localize anomalies responsible for liver segmentation model failures (e.g., metal artifacts and ascites) and identify MIDRC chest radiographs that deviate from those in ChestX-ray14. The overarching goal of this study is to demonstrate how generative modeling-based anomaly detection can improve interpretability in AI failure detection systems and support scalable data curation for large medical imaging repositories.

## 2. Materials and Methods

### 2.1. Data

Baseline training, baseline testing, and anomalous testing datasets were created for each application. In failure detection, “anomalous” denotes deviations from a baseline CT distribution that cause liver segmentation failures. For curation, it indicates attribute deviations from a baseline radiograph distribution.

#### 2.1.1. Failure Detection Datasets

For the failure detection task, the baseline datasets were derived from 206 abdominal CT scans (129 patients, 43% female, median age 64, acquired 2000–2017) from Anderson et al. [8] and 3029 CT abdominal scans (301 patients, 46% female, median age 63, acquired 2001–2002) from Woodland et al. [1], which focus on liver segmentation and StyleGAN2 benchmarking, respectively. A total of 254 scans from 50 randomly selected patients were withheld for baseline testing; the rest were used for training. Anomaly test datasets included 48 intraoperative abdominal CT scans (39 patients, 28% female, median age 59, acquired 2020–2022) from an ongoing liver ablation trial [13] (“Needles”), 33 abdominal CT scans (33 patients, 42% female, median age 66, acquired 2014–2023) with ascites (“Ascites”), 10 head and neck CT scans (10 patients, 40% female, median age 56, acquired 2001–2011) split axially into three datasets (“Brain”, “Lung”, and “Head and Neck”—slices containing neither the brain nor lungs), and 10 female pelvic CT scans (“Cervix”; 10 patients, 100% female, median age 38, acquired 2019). Table A1 (Appendix A) summarizes the demographics and clinical characteristics of the 3339 scans (525 patients). 

Needles and Ascites were included as metal artifacts and ascites have caused liver segmentation model failures [8]. Needles were of particular interest for metal artifact evaluation due to the role autosegmentation plays in minimal ablative margin assessment [13]. Non-liver datasets evaluated the detection of deviations from the intended model scope. All scans were retrospectively acquired from The University of Texas MD Anderson Cancer Center. All datasets were constructed such that there is no patient overlap across datasets.

The scans were all obtained in the Digital Imaging and Communications in Medicine (DICOM) format. Voxel values were windowed with a level of 50 and a width of 350, consistent with the default liver viewing settings in RayStation v10 (RaySearch Laboratories, Stockholm, Sweden), and mapped to the range [0, 255]. 2D axial slices were extracted for training and evaluation to enable both computational feasibility and effective anomaly detection. Training on full 3D volumes would have exceeded available GPU memory, necessitating a reduction in dimensionality. The anomalies of interest—needles and ascites—are clearly visible in individual 2D slices, justifying slice extraction as the form of dimensionality reduction. Moreover, these anomalies are small relative to the entire volume, so extracting slices protects against the possibility that their reconstruction errors could be masked by the substantial contextual information and noise inherent in 3D volumes.

These slices were converted to Portable Network Graphics (PNGs) using OpenCV (Python package version 4.10.0.84), de-identifying the images by removing header information, following the Health Insurance Portability and Accountability Act (HIPAA)’s Safe Harbor method. The final training dataset had 136,908 512 × 512 images, while the baseline test encompassed 17,037. A total of 250 slices were manually selected for the non-liver datasets, while 150 were selected for Needles and Ascites. Only slices clearly containing the target anatomy, pathology, or object were included in the anomaly datasets. These slices were selected under the guidance of a radiologist with 7 years of experience. Figure 1 displays images from all CT datasets.

#### 2.1.2. Data Curation Datasets

For the data curation task, 112,120 chest radiographs (30,805 patients, 46% female, median age 47, acquired 1992–2015) from the ChestX-ray14 dataset [12] constituted the baseline training dataset. All available chest radiographs (64,373) were downloaded from MIDRC in November 2022, with 1000 selected for baseline testing. Each baseline image was frontal view (either anteroposterior or posteroanterior) with visible ribs, entire lungs, no preprocessing, and the head at the top. Anomaly test datasets were created from the remaining radiographs—“Bone Suppression” (ribs hidden), “Filtered” (edge-filtered), “Missing Lung” (at least half of one lung absent), “Inverted” (grayscale-inverted), “No Anatomy”, and “Orientation” (head not at the top)—with up to 250 radiographs manually selected. To our knowledge, this is the first study to apply out-of-distribution detection to chest radiographs from MIDRC.

Table A2 (Appendix A) describes the demographics and clinical characteristics of the 2036 radiographs selected for testing. The test datasets primarily comprised computed radiographs, except for Orientation, which contained mostly digital radiographs. Of the test radiographs with pertinent metadata, 45% were from female patients (median age 55). The final Bone Suppression, Filtered, Missing Lung, and Inverted datasets contained 250 images, while No Anatomy and Orientation contained 13 and 23 images, respectively. The No Anatomy dataset contained solid-colored images, images with copious amounts of noise obscuring the anatomy, and an exam protocol. All images were of a 512 × 512 resolution. Figure 2 displays examples from all radiography datasets.

The ChestX-ray14 dataset [12] was downloaded from https://nihcc.app.box.com/v/ChestXray-NIHCC on 13 May 2022, in PNG format. These images did not have burned-in protected health information, and the algorithms were not shown the associated metadata, thereby satisfying HIPAA’s Safe Harbor de-identification method. Each radiograph was downsampled to a resolution of 512 × 512 using Pillow with a bicubic resampler (Python package version 11.1.0). The MIDRC radiographs were downloaded de-identified [14] from https://www.midrc.org/ on 11 October 2022, in DICOM format and downsampled to a resolution of 512 × 512 with OpenCV using a bilinear interpolator (Python package version 4.10.0.84). Pixels were rescaled linearly to the range [0, 255] using SimpleITK (Python package version 2.4.1) and were converted to 8-bit unsigned integers with NumPy (Python package version 1.26.4). The radiographs were subsequently saved as PNGs using OpenCV (Python package version 4.10.0.84).

### 2.2. Generative Modeling for Anomaly Detection

Our pipeline, presented in Figure 3, employs generative modeling for anomaly detection, i.e., the task of identifying inputs with attributes that deviate from a baseline distribution. For each application, a generative model was trained on the associated baseline training dataset. Each model was then used to reconstruct the test datasets. The error between the test images and their associated reconstructions was subsequently used as the anomaly detection score. The intuition behind this pipeline is that the generative models should struggle to reconstruct anomalous information, thereby producing higher scores for the anomalous images than for baseline images.

#### 2.2.1. Generative Model Training

A StyleGAN2 network [15] was trained per application using transfer learning from the Flickr-Faces-HQ dataset [16] and adaptive discriminator augmentation [17], both of which were shown to stabilize training and improve perceptual quality in prior work using StyleGAN2 to generate medical images [1]. Each was trained for 25,000 kimg (one kimg represents 1000 images shown to the network—approximately 164 epochs for failure detection and 223 epochs for data curation), with weights saved every 200 kimg (approximately every epoch), and those with the lowest Fréchet Inception Distance (FID) [18] selected.

Generative adversarial networks (GANs) [19] are an adversarial neural network framework where two networks, the generator and the discriminator, compete in a zero-sum game. The generator produces synthetic images, while the discriminator distinguishes between real and synthetic images. StyleGAN2 is a GAN where the generator progressively increases the resolution of a learned constant (4 × 4 × 512 for 512 × 512 output) through a synthesis network composed of 3 × 3 convolutional layers. Each layer is modulated by a style vector derived from a mapped latent code and injected with random noise to induce stochastic variation. The discriminator mirrors this progressive structure, evaluating images at multiple resolution scales. A schematic of the overall StyleGAN2 architecture is provided in Figure A1 (Appendix B). For 512 × 512 images, the mapping network consists of eight fully connected layers with approximately 2 million parameters. Both the synthesis and discriminator networks contain fifteen convolutional layers and approximately 28 million trainable parameters each. 

The official PyTorch implementation of StyleGAN2-ADA (StyleGAN2-ADA—Official PyTorch implementation. https://github.com/NVlabs/stylegan2-ada-pytorch. Accessed on 27 November 2023) was used with default parameters, apart from changing $\beta_{0}$ to 0.9 in the Adam optimizer and disabling mixed precision to stabilize training. Both horizontal flipping and adaptive discriminator augmentation were enabled for data augmentation. These hyperparameters were used based on prior evidence of their robustness across medical imaging domains [1]. The networks were trained on DGXs with either eight 40GB A100 or eight 80GB H100 GPUs, accessed using Kubernetes. It took approximately 4 days on A100s or 2.5 days on H100s to complete each training phase.

#### 2.2.2. Generative Modeling Evaluation

Generative performance was evaluated quantitatively using FID, Fréchet SwAV Distance (FSD) [20], and Fréchet Radiomics Distance (FRD) [21], which assess feature-wise similarity between real and generated distributions. FID is a widely used metric for generated image quality [22], while FSD and FRD are proposed improvements. These metrics compute the Fréchet Distance (FD), defined as$$d^{2}\left(\Sigma_{1},\Sigma_{2},\mu_{1},\mu_{2}\right)={\left|\mu_{1}-\mu_{2}\right|}^{2}+\mathrm{tr}\left(\Sigma_{1}+\Sigma_{2}-2{\left(\Sigma_{1}\Sigma_{2}\right)}^{\frac{1}{2}}\right),$$
between Gaussian distributions of real $\left(\Sigma_{R},\mu_{R}\right)$ and generated $\left(\Sigma_{G},\mu_{G}\right)$ features ($\mu_{i}$ and $\Sigma_{i}$ represent mean and covariance). FID and FSD leverage ImageNet-trained [23] InceptionV3 [24] and SwAV (Swapping Assignments between Views) [25] features, respectively, while FRD employs domain-specific radiomic features. Despite relying on ImageNet, both FID and FSD have been shown to align with expert assessments of generated medical image realism, with FSD achieving a statistically significant correlation [26]. To provide context, FDs were computed between training dataset halves, including calculations where one half contained uniformly applied Gaussian noise or blur (Figure A2, Appendix B). Means and standard deviations (SDs) were calculated across five image generation or dataset halving repetitions. These metrics were complemented by qualitative evaluations of anatomical plausibility and image fidelity by a radiologist with seven years of experience.

#### 2.2.3. Image Reconstruction

Test images were reconstructed by optimizing input vectors to the trained StyleGAN2 mapping networks via backpropagation, thereby minimizing the differences between the generated (i.e., reconstructed) and original images [27]. Encoders were not used to prevent anomalies from leaking into reconstructions [28]. Reconstructions were scored by comparing the reconstructed images to their test image counterparts using mean squared error (MSE) and Wasserstein distance (WD), the latter chosen for its spatial invariance. Reconstruction quality was evaluated qualitatively by analyzing the reconstructions with the highest and lowest scores per dataset (for both MSE and WD). On CT only, whole-image scoring was compared to scoring within the body to minimize the influence of surrounding pixels.

Images were reconstructed using the projector.py file from the official StyleGAN2-ADA (PyTorch) repository (StyleGAN2-ADA—Official PyTorch implementation. https://github.com/NVlabs/stylegan2-ada-pytorch. Accessed on 27 November 2023). The file was updated so that the reconstructions were saved in grayscale. Backpropagation was performed with the default 1000 steps. A default random seed was used for the anomaly detection evaluation. For the proportion of patches highlighting the evaluated anomaly calculation, backpropagation was performed with random seeds 0–4.

#### 2.2.4. Anomaly Detection

Anomaly detection performance was evaluated with the area under the receiver operating characteristic curve (AUROC) using scikit-learn. It was computed between anomaly scores from each anomalous test dataset and its corresponding baseline test dataset. Instead of setting a score threshold to differentiate the baseline and anomaly classes, AUROC provides a comprehensive measure of performance by evaluating all thresholds. To address class imbalance and estimate performance variability, repeated random subsampling was performed by drawing images without replacement from the baseline test datasets to match the size of the anomalous datasets. Mean AUROC values and their associated SDs were computed across 50 subsamples.

#### 2.2.5. Anomaly Localization

For local anomalies (i.e., needles and ascites), localization was performed by identifying the image region with the highest reconstruction error. This approach offers a form of interpretability for failure detection pipelines by highlighting the image region on which a model is most likely to fail. Localization performance was evaluated by measuring the proportion of top-scoring patches that contained the target anomaly. These proportions were calculated for patch sizes d×d where d ϵ {32, 64, 128}. For each configuration, means and SDs were computed across five reconstructions generated with different random seeds. The proportion of highest-scoring patches containing the evaluated anomaly included strong imaging artifacts caused by a needle as a successful localization of the needle.

### 2.3. Statistical Analysis

One-sided permutation tests (significance level α = 0.05) were used to compare FDs between generated and manipulated images (sample size n = 30), AUROCs between reconstruction metrics (n = 50), AUROCs between whole-image and body-only scoring (n = 50), and proportions across scoring functions and patch sizes (n = 5). Simulation-based power analyses confirmed ≥80% power for all sample sizes.

Permutation test assumptions of exchangeability under the null hypothesis and independence of observations were satisfied for all analyses. For FD comparisons, each test statistic was computed from independently and identically sampled subsets. For AUROC and proportion comparisons, subpopulations were derived from matched data splits, ensuring that permutations preserved the joint distribution under the null. Subsamples were drawn independently to maintain observation-level independence.

To assess statistical power, we conducted simulation-based power analyses using a permutation testing framework. For each sample size, we simulated 1000 datasets by drawing samples from normal distributions matched to the observed means and standard deviations of the two groups. For each simulated dataset, we applied a one-sided permutation test with 1000 iterations to evaluate the null hypothesis of no difference between groups. Power was estimated as the proportion of simulations in which the test yielded a *p*-value below the significance threshold of α = 0.05.

### 2.4. Code Availability

All code is available on GitHub at https://github.com/mckellwoodland/gan_anom_detect (accessed 10 May 2024 through 13 October 2025). The chest radiography StyleGAN2 weights are available on Zenodo at https://zenodo.org/records/14901472 [29] (accessed on 20 February 2025). 

## 3. Results

### 3.1. Evaluation of Generated Image Quality

To assess the realism of images generated by the StyleGAN2 models, we computed Fréchet Distances (FDs) between the training and generated distributions. As shown in Table 1, the generated images were significantly more similar to the training data than the real images with added Gaussian noise or blur were (*p* < 0.001, permutation tests on FDs), with FDs reduced by 69%, on average. This indicates high fidelity in the generative models.

Figure A3 (Appendix B) illustrates representative, randomly sampled images generated by the abdominal CT and chest radiography models, which a radiologist qualitatively evaluated. The abdominal CT model successfully synthesized axial slices spanning the liver’s inferior to superior edges, capturing varying levels of noise and organs with and without contrast, and producing fine-grained details such as ablated liver tissue. However, some images exhibited reduced contrast detectability and blurred tissue interfaces. The chest radiography model generated anatomically accurate frontal-view radiographs with varying exposure levels but struggled to reproduce details such as central lines, surgical clips, and embedded text. 

### 3.2. Reconstruction Performance and Interpretation

To assess the reconstructive fidelity of the StyleGAN2 models, we present representative reconstructions with the lowest and highest reconstruction errors for each test dataset in Figure 4 (abdominal CT) and Figure 5 (chest radiography). These visualizations highlight the models’ ability to preserve baseline anatomical structures, while revealing their limitations in handling novel components—a desirable trait for anomaly detection tasks.

In the abdominal CT model (Figure 4), reconstructions of the baseline liver images were fairly anatomically consistent. When faced with non-liver inputs, the model attempted to generalize by adapting abdominal anatomy. For instance, it filled the abdominal cavity with liver-like tissue to represent the brain. This led to poor reconstructions of unfamiliar anatomical parts, including the hips, shoulders, and jaw. Most notably, the model completely failed to reconstruct the anomalies under evaluation—needles and ascites—leading to their downstream detection.

Similarly, the chest radiography model (Figure 5) effectively reconstructed baseline structures and handled radiographs with removed components, including cases with bone suppression, partially visible lungs, and the complete absence of anatomy. However, the model struggled with novel objects—such as surgical clips and scissors—as well as unseen image processing techniques, including edge filtering, grayscale inversion, and rotated orientations. These limitations reinforce the model’s potential utility in anomaly detection pipelines, where failure to reconstruct unfamiliar features can serve as a signal for abnormality.

### 3.3. Quantitative Anomaly Detection Performance

Table 2 and Table 3 summarize the AUROC values obtained from reconstruction-based anomaly scoring, applied to whole images and regions restricted to the human body, respectively. Across all test datasets, a mean AUROC of 0.86 ± 0.13 was achieved, indicating strong performance. Notably, WD outperformed MSE on the CT datasets, whereas MSE outperformed WD on the radiography datasets (*p* < 0.001, permutation tests comparing AUROCs), with the exception of the No Anatomy dataset. Restricting scoring to the human body significantly improved performance compared to whole image scoring (*p* < 0.001, permutation tests comparing AUROCs), with the exception of the Needles dataset. This approach yielded a mean improvement of 43% ± 60%, underscoring the benefit of focusing on anatomically relevant regions.

Among the non-liver datasets, the Brain dataset achieved the highest AUROC (1.00 ± 0.00), while the Cervix dataset had the lowest (0.90 ± 0.01), reflecting high performances (Table 3). For the radiographs, the highest AUROC (0.98 ± 0.00) was observed on the Filtered dataset (Table 2), where the model failed to reconstruct the edge filtering (Figure 5). Conversely, the lowest AUROC (0.63 ± 0.04) was recorded on the No Anatomy dataset. Although the model failed to reconstruct protocols and images with excessive noise, it was able to reconstruct entirely black images (6 of 13 in No Anatomy), which contributed to the reduced AUROC (Figure 5). Overall, the methodology was most effective at detecting anomalies involving the inclusion of novel information (e.g., Filtered, Inverted), rather than those involving the exclusion or suppression of expected content (e.g., Bone Suppression, No Anatomy).

### 3.4. Localization of Anomalous Regions

Table 4 details the localization results for needles and ascites. The combination of WD with a patch size of 128 × 128 achieved the best performance (*p* < 0.001, permutation tests comparing proportions), successfully localizing 70% ± 3% of the needles and 93% ± 1% of ascites. Figure 6 displays representative examples of anomalies accurately localized within the 128 × 128 patches exhibiting the highest WD, alongside cases where localization failed. The generative model frequently replaced needles and ascites with normal liver tissue, leading to elevated WD values in patches containing these anomalies. Localization errors that arose were primarily attributed to underrepresented components—such as tumors, calcifications, and contrast-enhanced adjacent organs—being correctly identified as anomalies, as well as suboptimal reconstructions of regions with high-intensity structures (e.g., spines and ribs).

## 4. Discussion

This study applied generative modeling to the identification and localization of anomalous information and has several key findings. First, generative models effectively localized anomalies that caused a liver segmentation model to fail. Second, generative models identified chest radiographs that contained information that was not present in a specified distribution. Third, the proposed WD was more effective than MSE at reconstruction scoring on abdominal CT data.

Generative modeling localized anomalies that have caused an AI model to fail, supporting the methodology’s application in clinical settings, where it could introduce interpretability into failure detection pipelines. While other methods, such as feature-based out-of-distribution detection, can accurately and efficiently detect model failures [9], they are uninterpretable. When these methods detect model failure for a given image, generative modeling can localize the image region that differs the most from what the model encountered during training. This localization provides interpretability by identifying the image region that was most likely to cause the failure detection. Some out-of-distribution detection methods, such as Monte-Carlo Dropout [30], can provide interpretability by overlaying uncertainty maps onto the original images. The problem with this form of interpretability is that AI models express high certainty, even when they fail on anomalous information [31]. In contrast, generative modeling decouples this interpretability from the poorly calibrated certainty of AI models.

In addition, generative modeling identified radiographs that deviated from a baseline distribution due to the inclusion of anomalies. This finding aligns with research by Nakao et al. [32], which demonstrated that generative modeling could distinguish “No Opacity/Not Normal” and “Opacity” chest radiograph distributions from a “Normal” distribution, achieving AUROCs of 0.70 and 0.84, respectively. Traditionally, generative modeling has been used in medical imaging for disease detection [4], but our study uses it to identify novel imaging characteristics, including unseen processing techniques and viewing parameters. These imaging characteristics can be identified by comparing the original and reconstructed images with the highest reconstruction scores and discriminating the largest discrepancy between them. By detecting these unseen characteristics, we propose that generative modeling can assist in data curation for large repositories by identifying images that fall outside the scope of the repositories, such as the imaging protocol identified in this study.

Our final key finding was that the proposed WD outperformed MSE for reconstruction scoring on CT. The CT data exhibited greater variability in intensity distributions compared to the chest radiographs, which may have contributed to WD’s superior performance. Related literature has indicated that metrics such as Structural Similarity Index [33] and Learned Perceptual Image Patch Similarity [34] are also effective for reconstruction assessments. While we are the first to use WD as a reconstruction score, other research has demonstrated its utility in other aspects of anomaly detection. For instance, Cheng et al. utilized WD with spatial filtering to detect anomalous targets in hyperspectral images [35].

This study has several limitations. First, baseline data were separated from anomalous data based on one differing attribute (such as comparing abdominal CTs with and without needles). While this clear separation enabled algorithmic evaluation, non-delineated underrepresented attributes within the baseline datasets adversely affected the reported results when identified as anomalous. Consequently, the reported results were worse than the true detection performance of underrepresented attributes. 

Second, the reported results, which were obtained through an unsupervised methodology, may not be as competitive as those derived from supervised approaches. Despite this potential for performance loss on predefined anomalies, unsupervised techniques offer the advantage of detecting all deviations from a baseline distribution. This capability is essential for deployment since model creators and repository curators may not be aware of all the anomalies their methodology might encounter in practice. Additionally, unsupervised methods alleviate the need for extensive data curation.

Finally, the StyleGAN2 model used in this study has several weaknesses. First, it is too computationally intensive for 3D modeling in high resolutions, necessitating anomaly detection to operate in a 2D context. While this may limit certain volumetric applications, 2D modeling was appropriate for this work, as the evaluated 3D anomalies—needles and ascites—were clearly visible in 2D slices and small enough that 3D context could have obscured their poor reconstructions with excess noise. If future applications require volumetric modeling, patch-based 3D alternatives may offer a computationally feasible path forward. Second, generative performance may have been improved by fine-tuning the StyleGAN2 architecture or by using a more contemporary generative model, such as diffusion models. Third, StyleGAN2 lacks the inherent ability to reconstruct images, requiring the use of backpropagation, which may hinder anomaly localization by poorly reconstructing normal regions when it is too focused on anomalies. 

Despite these limitations in generative modeling, this work establishes a baseline for employing generative modeling in two underexplored but critical tasks: (1) anomaly localization in failure detection systems and (2) the identification of out-of-scope images submitted to large repositories. While the current implementation using StyleGAN2 is not yet suitable for clinical deployment, the results demonstrate that generative models consistently fail to reconstruct anomalous features, enabling their detection and spatial localization. This behavior offers a pathway toward interpretable failure detection, which is essential for safe and transparent AI integration into clinical workflows. Similarly, the ability to flag radiographs with unseen attributes suggests a scalable approach to repository intake quality control. By framing these tasks and providing a quantitative benchmark, our study lays the groundwork for future research applying and evaluating generative modeling approaches for these specific applications.

## 5. Conclusions

This study demonstrates that generative modeling-based anomaly detection has strong potential for improving interpretability and scalability in medical imaging AI systems. We showed that StyleGAN2 models failed to reconstruct anomalous features such as needles and ascites, enabling their localization with rates up to 93%. These results support the use of generative modeling to localize anomalies that have caused AI model failures, offering a pathway toward interpretable failure detection. Additionally, the models identified radiographs with out-of-distribution attributes (e.g., filtering, inversion, missing anatomy), achieving AUROCs up to 0.98, which suggests applicability to streamlining data repository intake by detecting out-of-scope submissions. Finally, the proposed Wasserstein distance metric significantly outperformed mean squared error on CT data (*p* < 0.001), offering a robust alternative for reconstruction-based anomaly scoring. These findings collectively support further exploration of generative modeling methodologies for the proposed tasks of interpretable clinical AI failure detection and large-scale data curation.

## Figures and Tables

**Figure 1 bioengineering-12-01106-f001:**
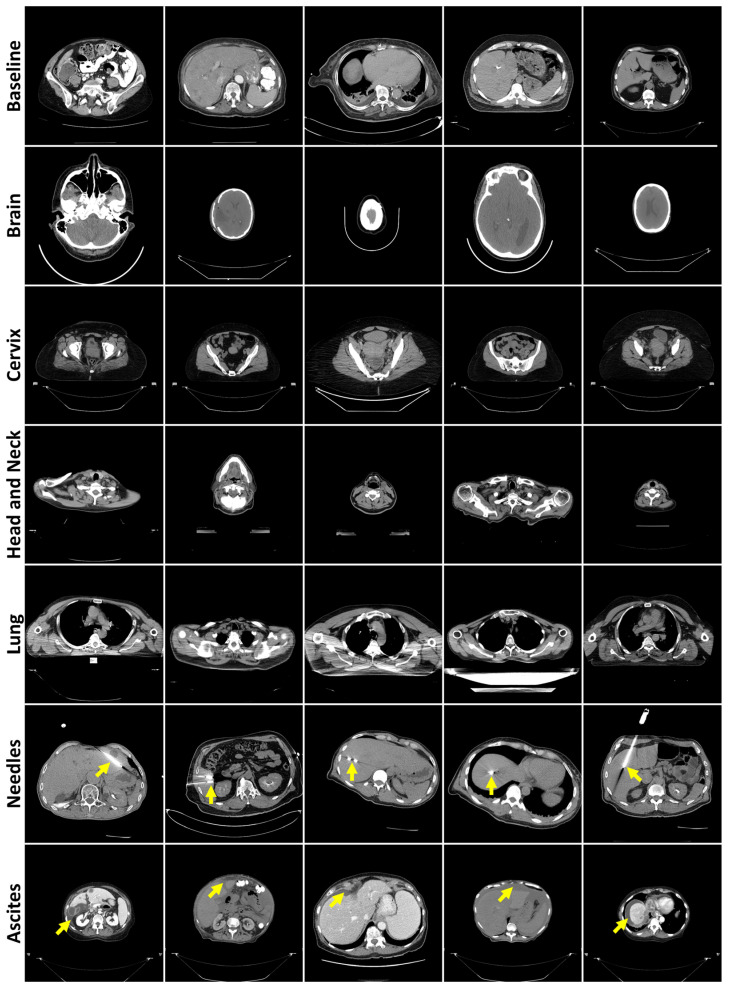
Representative axial CT slices from each failure detection test dataset, separated by rows. Yellow arrows highlight the presence of needles (row 6) or ascites (row 7).

**Figure 2 bioengineering-12-01106-f002:**
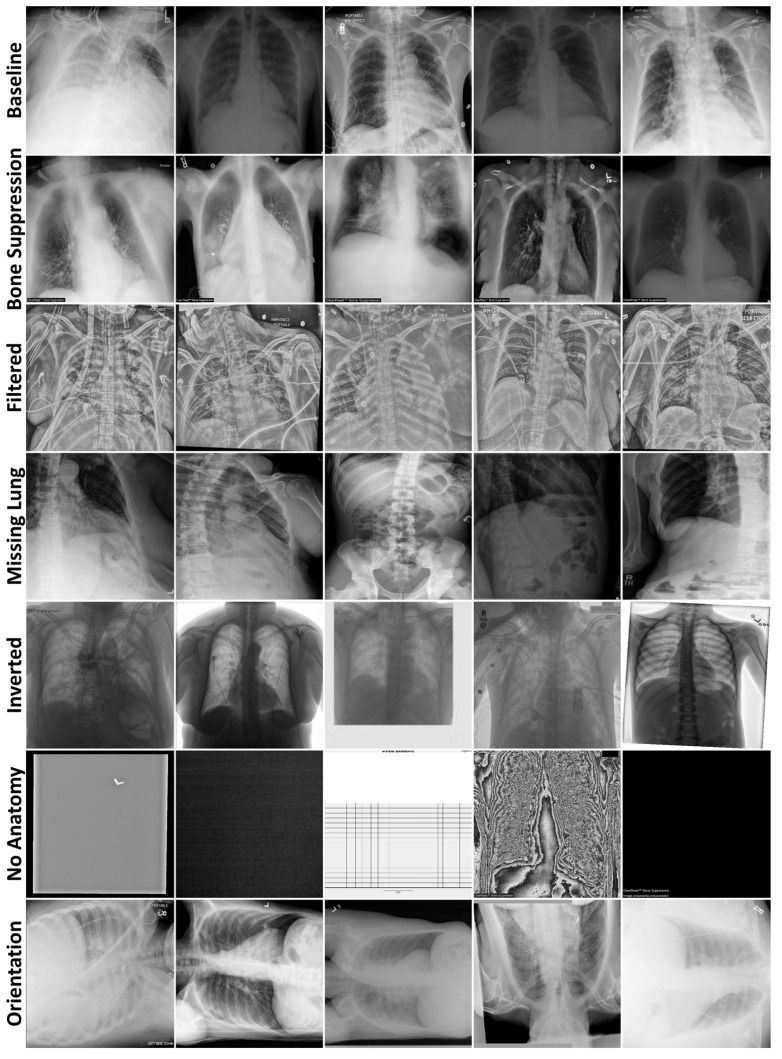
Example radiographs from each data curation test dataset, separated by rows.

**Figure 3 bioengineering-12-01106-f003:**
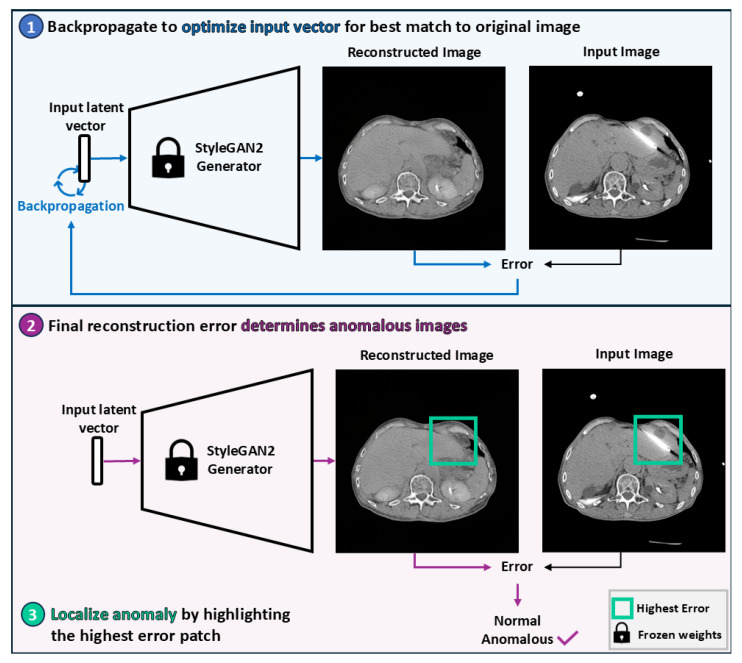
Overview of the StyleGAN2-based anomaly detection pipeline. A latent vector is optimized via backpropagation to produce a reconstruction that most closely resembles the original image. The magnitude of the final reconstruction error determines whether the image is considered anomalous. Anomalies are localized by identifying the patch with the highest reconstruction error.

**Figure 4 bioengineering-12-01106-f004:**
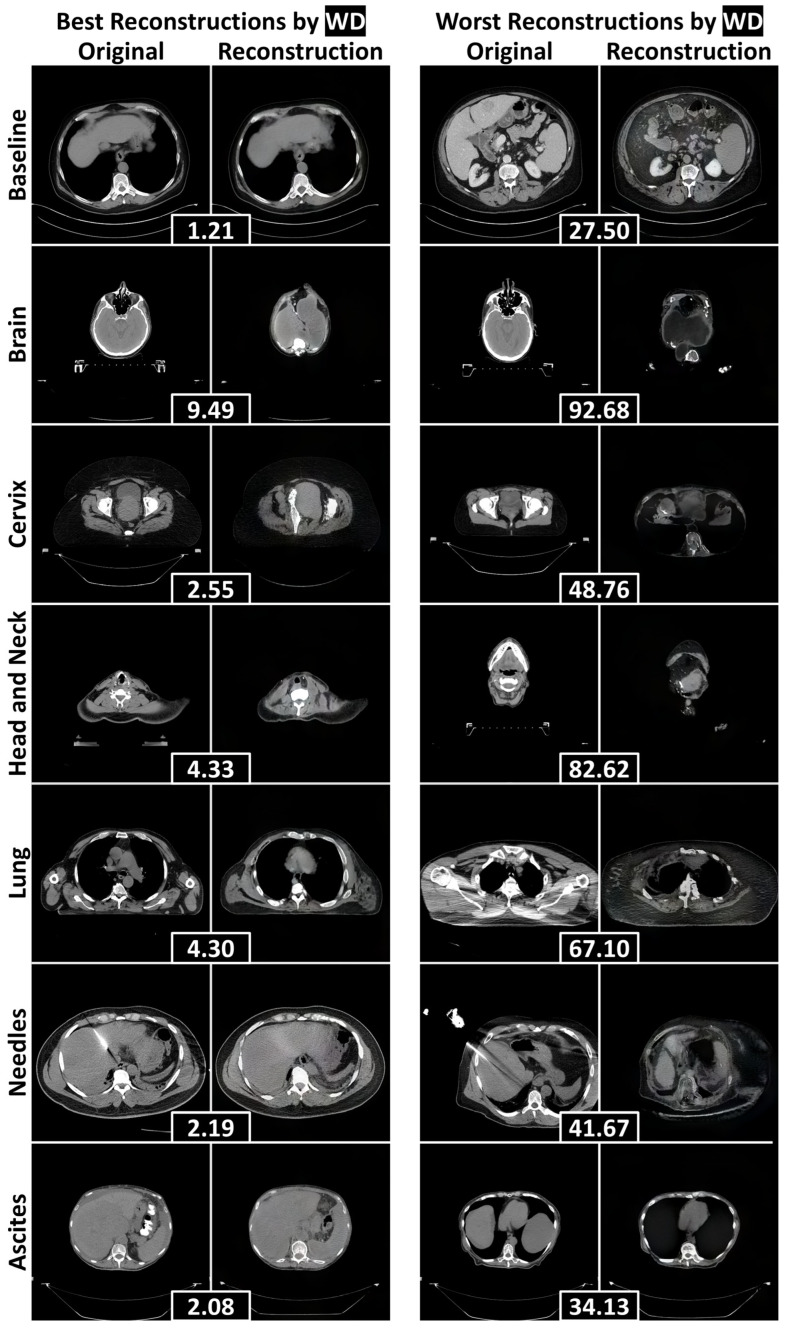
The best and worst reconstructions (according to the WD calculated within the human body) for each failure detection test dataset, separated by rows. WDs between an original axial abdominal CT slice and its reconstruction are included in black boxes between the pair. Abbreviations: Wasserstein distance (WD).

**Figure 5 bioengineering-12-01106-f005:**
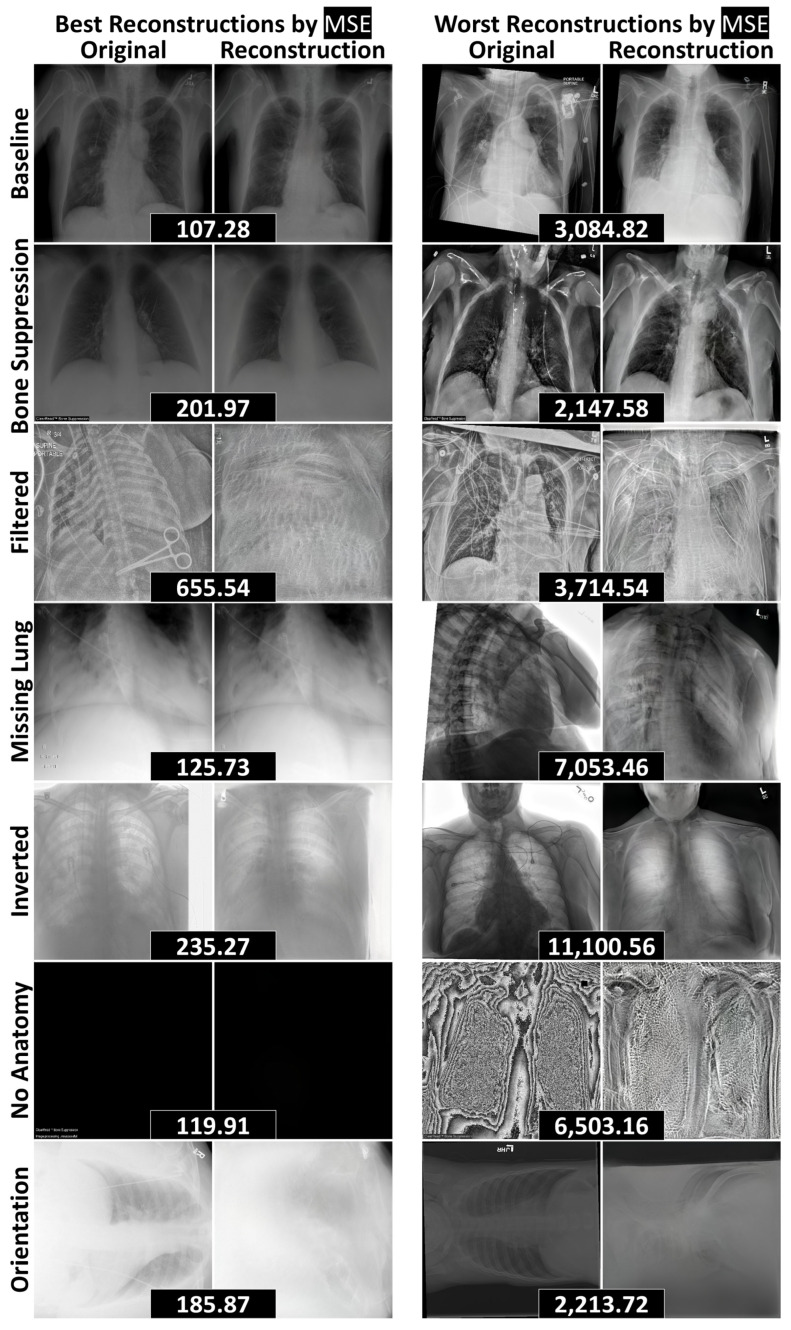
The best and worst reconstructions (according to MSE) for each data curation test dataset, separated by rows. MSEs between an original radiograph and its reconstruction are included in black boxes between the pair. Abbreviations: mean squared error (MSE).

**Figure 6 bioengineering-12-01106-f006:**
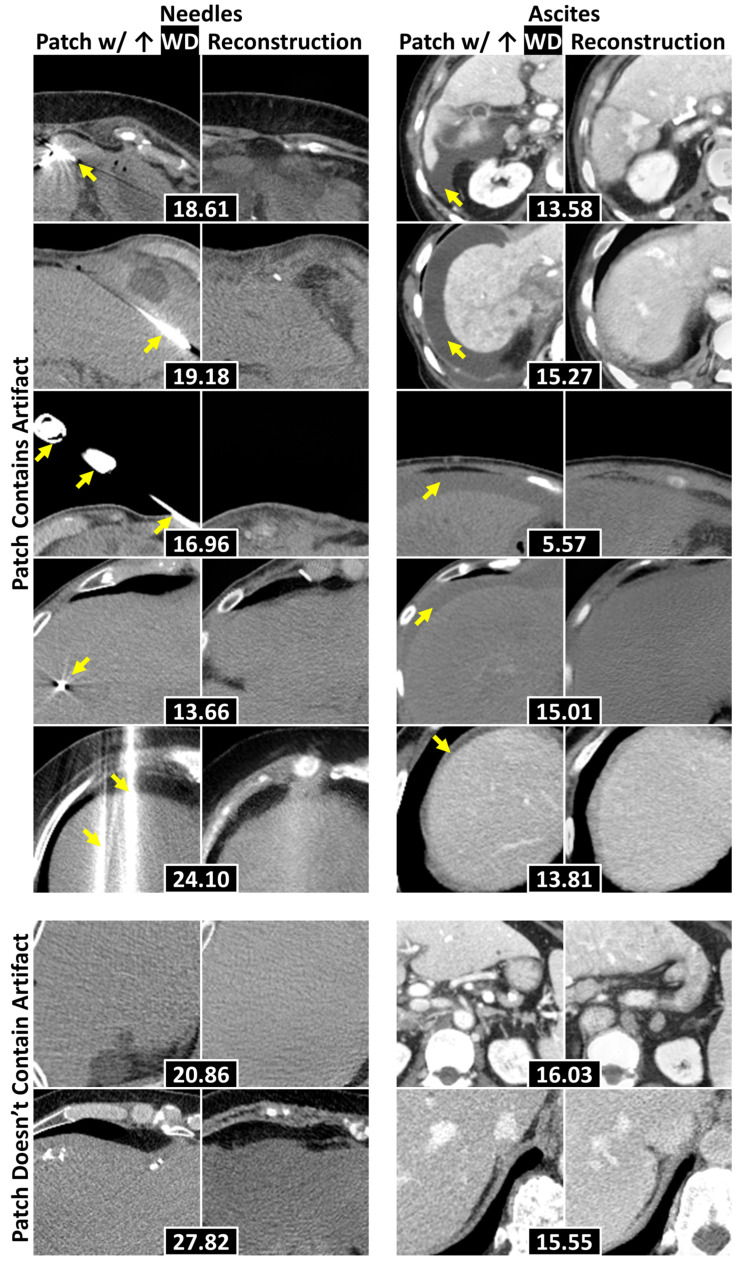
Example 128 × 128 patches of axial abdominal CT slices with their corresponding reconstructions. Each pair attained the highest WD within a slice, with WDs placed in black boxes. The top five rows display examples where the evaluated anomalies (e.g., needles and ascites—separated by columns) were localized, with yellow arrows pointing to the anomalies. The bottom two contain examples where the anomalies were not localized. Abbreviations: Wasserstein distance (WD).

**Table 1 bioengineering-12-01106-t001:** Fréchet Distances. Mean (±SD) FDs, which measure the similarity of two distributions. Baseline FDs compared two halves of the training dataset. Generated FDs compared generated images to the training dataset. Noise and Blur FDs compared two halves of the training dataset, where one half had either Gaussian noise or blur applied. * refers to *p*-values from one-sided permutation tests comparing generated and manipulated FDs. Arrows denote that lower is better. Abbreviations: Fréchet Inception Distance (FID), Fréchet SwAV distance (FSD), Fréchet Radiomics Distance (FRD), standard deviation (SD), and Fréchet Distance (FD).

Dataset		FID (±SD) ↓	FSD (±SD) ↓	FRD (±SD) ↓
Liver CT	Baseline	0.26 (±0.00)	0.01 (±0.00)	0.01 (±0.00)
	Generated	3.37 (±0.05) -	0.96 (±0.00) *p* < 0.001 *	0.81 (±0.04) -
	Noise	31.43 (±0.17) *p* < 0.001 *	4.45 (±0.01) *p* < 0.001 *	216.28 (±0.34) *p* < 0.001 *
	Blur	47.65 (±0.13) *p* < 0.001 *	0.44 (±0.00) -	164.54 (±0.60) *p* < 0.001 *
Chest Radiography	Baseline	0.40 (±0.00)	0.02 (±0.00)	0.02 (±0.01)
	Generated	4.49 (±0.03) -	0.92 (±0.00) -	8.56 (±0.03) -
	Noise	86.65 (±0.19) *p* < 0.001 *	14.90 (±0.01) *p* < 0.001 *	403.29 (±10.57) *p* < 0.001 *
	Blur	36.98 (±0.12) *p* < 0.001 *	1.41 (±0.01) *p* < 0.001 *	49.64 (±5.30) *p* < 0.001 *

**Table 2 bioengineering-12-01106-t002:** Anomaly Detection Results on Full Images. Mean AUROCs (±SD) across 50 random subsamples for each anomaly dataset and reconstruction metric. * refers to *p*-values from one-sided permutation tests comparing WD- and MSE-based AUROCs. Arrows denote that higher is better. Abbreviations: Wasserstein distance (WD), area under the receiver operating characteristic curve (AUROC), mean squared error (MSE), and standard deviation (SD).

Dataset		WD-Based AUROC (±SD) ↑	MSE-Based AUROC (±SD) ↑	*p*
Failure Detection	Brain	0.66 (±0.03)	0.20 (±0.02)	*p* < 0.001 *
	Cervix	0.71 (±0.02)	0.48 (±0.02)	*p* < 0.001 *
	Head and Neck	0.37 (±0.02)	0.15 (±0.01)	*p* < 0.001 *
	Lung	0.89 (±0.01)	0.79 (±0.01)	*p* < 0.001 *
	Needles	0.69 (±0.02)	0.58 (±0.03)	*p* < 0.001 *
	Ascites	0.60 (±0.02)	0.43 (±0.03)	*p* < 0.001 *
Data Curation	Bone Suppression	0.68 (±0.02)	0.79 (±0.01)	*p* < 0.001 *
	Filtered	0.57 (±0.02)	0.98 (±0.00)	*p* < 0.001 *
	Missing Lung	0.58 (±0.02)	0.82 (±0.01)	*p* < 0.001 *
	Inverted	0.84 (±0.01)	0.90 (±0.01)	*p* < 0.001 *
	No Anatomy	0.63 (±0.04)	0.62 (±0.01)	*p* = 0.013 *
	Orientation	0.74 (±0.05)	0.78 (±0.03)	*p* < 0.001 *

**Table 3 bioengineering-12-01106-t003:** Anomaly Detection Results Within the Human Body (CT). Mean AUROCs (±SD) across 50 random subsamples for each failure detection anomaly dataset and reconstruction metric, calculated only within the human body. † denotes that the AUROC is significantly higher than the corresponding AUROC calculated on the full images. * refers to *p*-values from one-sided permutation tests comparing WD- and MSE-based AUROCs. ** refers to *p*-values from one-sided permutation tests comparing AUROCs calculated within the human body to those calculated on full images. Arrows denote that higher is better. Abbreviations: Wasserstein distance (WD), area under the receiver operating characteristic curve (AUROC), mean squared error (MSE), and standard deviation (SD).

Dataset		WD-Based AUROC (±SD) ↑	MSE-Based AUROC (±SD) ↑	*p*
Non-liver	Brain	1.00 (±0.00) ^†^ *p* < 0.001 **	0.90 (±0.01) ^†^ *p* < 0.001 **	*p* < 0.001 *
	Cervix	0.90 (±0.01) ^†^ *p* < 0.001 **	0.70 (±0.02) ^†^ *p* < 0.001 **	*p* < 0.001 *
	Head and Neck	0.96 (±0.00) ^†^ *p* < 0.001 **	0.90 (±0.01) ^†^ *p* < 0.001 **	*p* < 0.001 *
	Lung	0.94 (±0.01) ^†^ *p* < 0.001 **	0.90 (±0.01) ^†^ *p* < 0.001 **	*p* < 0.001 *
Liver Anomaly	Needles	0.69 (±0.02) *p* = 0.160 **	0.60 (±0.02) ^†^ *p* < 0.001 **	*p* < 0.001 *
	Ascites	0.67 (±0.02) ^†^ *p* < 0.001 **	0.50 (±0.03) ^†^ *p* < 0.001 **	*p* < 0.001 *

**Table 4 bioengineering-12-01106-t004:** Proportion of Identified Anomalies by Reconstruction Metric and Patch Size. The mean (±SD) proportion of highest-scoring patches that contain the evaluated anomaly (needles or ascites) across five reconstructions with different seeds. Proportions are reported for patch sizes d × d where d∈{32,64,128}. * refers to *p*-values from one-sided permutation tests comparing proportions. Arrows denote that higher is better. Abbreviations: Wasserstein distance (WD), mean squared error (MSE), and standard deviation (SD).

Dataset	WD ↑			MSE ↑		
	32	64	128	32	64	128
Needles	0.43 (±0.01) *p* < 0.001 *	0.41 (±0.02) *p* < 0.001 *	0.70 (±0.03)	0.44 (±0.01) *p* < 0.001 *	0.38 (±0.03) *p* < 0.001 *	0.35 (±0.03) *p* < 0.001 *
Ascites	0.55 (±0.06) *p* < 0.001 *	0.71 (±0.03) *p* < 0.001 *	0.93 (±0.01)	0.44 (±0.05) *p* < 0.001 *	0.66 (±0.04) *p* < 0.001 *	0.86 (±0.02) *p* < 0.001 *

## Data Availability

The radiographical datasets presented in this study are openly available in Box at https://nihcc.app.box.com/v/ChestXray-NIHCC/folder/36938765345 (ChestX-ray14) (accessed on 13 May 2022) and MIDRC at https://www.midrc.org/ (accessed on 11 October 2022). The CT datasets presented in this study are available upon request from the corresponding author, in accordance with IRB protocol. The datasets are not publicly available due to patient privacy. Requests to access the datasets should be directed to K.K.B.

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
