# Peer review of "Generative Modeling for Interpretable Anomaly Detection in Medical Imaging: Applications in Failure Detection and Data Curation"

_bioengineering, 2025, doi:10.3390/bioengineering12101106_

Round 1

Reviewer 1 Report

Comments and Suggestions for Authors

The authors develop generative modeling-based anomaly detection with regard to enhancing the interpretability of model failure detection results and identifying out-of-scope images submitted to large data repositories. Therefore, the authors use generative modeling-based anomaly detection to enhance interpretability in AI failure detection systems and to aid data curation for large repositories.

My following comments should be taken into consideration to revise the paper version.

  1. It would be better to describe the construction of neural networks, which are used in models. How many levels are there in networks? How many neurons are there in levels? How many parameters (total number of neurons) are there in models?
  2. What reason is there for such choice of the level and parameter numbers?

Reviewer 2 Report

Comments and Suggestions for Authors

The paper “aims to leverage generative modeling–based anomaly detection to enhance interpretability in AI failure detection systems and to aid data curation for large repositories”. Large data repositories are employed in the research, “pivotal for advancing medical AI research by providing widespread data access [10]”.

The work is interesting, but some important aspects remain to be properly pointed out in order to appreciate novelty and generalizability.

The study, in fact, has several/severe limitations, correctly recognized and addressed in rows 406-427, which not only prevent the use in a clinical practice, very far to come, but also make difficult decisions on the overall importance of what proposed. At the present, a certain number of papers exist, dealing with the topics (keywords) highlighted in the paper with applications in many fields of research. With respect to the current state of the art, the improvements/ameliorations of what proposed do not emerge sufficiently clear. It seems that  “anomalies that caused failure in Anderson et al. [8]” have been resolved, or that “finding aligns with research by Nakao et al. [32]” ,  but these comments do not avoid the question on what has been added with respect to the literature. In my opinion, above all, a potential reader claims more precise comparisons and  numerical values to appreciate novelties, and not only a description too focused on what proposed.

Very intelligently, public datasets were chosen, as well as widely used metrics for performance evaluation, but we do not know if other researchers have used the same data for the same purposes and with what results. Even this is important for a constructive comparison.

Therefore, from the above, my suggestion is a reorganization in order to make more manifest these aspects with respect to the current literature.

Reviewer 3 Report

Comments and Suggestions for Authors

Line 107: “Two-dimensional axial slices were extracted for training and evaluation due to computational challenges in three-dimensional modeling.” The authors reduce inherently 3D tasks to 2D, undermining clinical relevance, but provide no quantitative justification or attempt at patch-based 3D.

Using default StyleGAN2-ADA hyperparameters without tuning for medical data is unjustified and weakens conclusions.

Reliance on ImageNet features for medical imaging is problematic. The authors admit the mismatch but do not provide stronger domain-specific evaluation metrics.

Simulation assumptions (normality of reconstruction scores, etc.) are not validated.

The variance is very high; ±0.13 suggests instability. The authors should analyze failure cases.

Round 2

Reviewer 2 Report

Comments and Suggestions for Authors

Improvements have been done.

Please, correct some repetitions (for instance, row 49),

equation in 230 is lacking of an exponent on the module.

Reviewer 3 Report

Comments and Suggestions for Authors

The manuscript has been sufficiently improved to warrant publication in Bioengineering

Author Response

Comments 1: The manuscript has been sufficiently improved to warrant publication in Bioengineering.

Response 1: Thank you for the time you took and the valuable feedback you provided when reviewing our manuscript.